# Myoma with Hypermenorrhea Treated with Ultrasound-Guided Microwave Ablation of the Inflowing Blood Vessels to the Uterine Myoma: A Case

Toshiyuki Kakinuma *, Takahumi Ohkusa, Takumi Shinohara, Ayano Shimizu, Rora Okamoto, Masataka Kagimoto, Ayaka Kaneko, Kaoru Kakinuma, Kaoru Yanagida, Nobuhiro Takeshima and Michitaka Ohwada

Department of Obstetrics and Gynecology, International University of Health and Welfare Hospital, Nasushiobara 329-2763, Japan
* Correspondence: tokakinuma@gmail.com; Tel.: +81-287-39-3060

**Abstract:** Microwave endometrial ablation (MEA) is a minimally invasive treatment for uterine myoma with hypermenorrhea, which can replace conventional hysterectomy. However, cases requiring additional treatment because of postoperative recurrence are often encountered. MEA cauterizes the endometrium and is not recommended for patients who wish to preserve fertility. We present the cases of a patient with myoma-related hypermenorrhea who underwent microwave ablation of the inflowing blood vessels to the uterine myoma under transvaginal ultrasound guidance. A 43-year-old woman was diagnosed with chronic myeloid leukemia and treated with dasatinib 2 years ago. Worsening hypermenorrhea was observed after treatment initiation. Ultrasound and pelvic magnetic resonance imaging revealed a uterine myoma. Therefore, she underwent MEA under transvaginal ultrasound guidance. Visual analog scale evaluation demonstrated considerable improvement in hypermenorrhea and dysmenorrhea; the myoma size showed reduction. The postoperative course was uneventful, and the patient was discharged on the day after surgery. No postoperative complications were observed. This patient is currently undergoing infertility treatment. The microwave ablation of myoma under transvaginal ultrasound guidance can effectively and safely reduce the myoma size. These findings suggest that this method is a novel treatment option for patients with myoma-related hypermenorrhea who wish to preserve their fertility and have children.

**Keywords:** uterine myoma; microwave; endometrial ablation; hypermenorrhea; dysmenorrhea

## 1. Introduction

Uterine myomas are benign gynecologic tumors that are often encountered in clinical practice. Uterine myomas lead to conditions, such as hypermenorrhea and algomenorrhea, resulting in severe anemia and hindering daily activities. These result in reduced quality of life for women and limit their social activities. Microwave endometrial ablation (MEA) aims to reduce the volume of menstruation or promote amenorrhea by the destruction of the endometrial basal layer, reducing the function through protein coagulation based on dielectric tissue heating due to microwave irradiation. MEA is a treatment for functional hypermenorrhea refractory to conservative treatment, hypermenorrhea associated with systemic diseases or therapeutic agents, and organic hypermenorrhea caused by uterine myoma and adenomyosis. MEA is a minimally invasive treatment that can be used as a substitute for conventional hysterectomy. MEA has also become popular in Japan because it has been covered by insurance since April 2012 as a minimally invasive treatment for hypermenorrhea. Since our hospital introduced MEA in January 2016, we have reported the efficacy of this treatment [1]. Although MEA can be expected to have a therapeutic effect on uterine fibroids, there are some cases wherein additional treatment is required because of postoperative recurrence [2,3], as it is often difficult to manage. Additionally, this treatment

method cauterizes the endometrium and is not recommended for patients who wish to preserve fertility. Uterine artery embolization (UAE) and magnetic resonance imaging (MRI)-guided focused ultrasound (MRgFUS) have recently been somewhat effective as conservative treatments. However, the treatment outcomes and complications are not always satisfactory compared with those associated with surgical treatment [4,5].

We herein introduce microwave ablation of the inflowing blood vessels to the uterine myoma under transvaginal ultrasound guidance as a new treatment method for patients with uterine myoma with menorrhagia who wish to preserve fertility.

## 2. Case Presentation

Case: A 43-year-old woman
History of pregnancy and delivery: One pregnancy and one delivery
Chief complaint: Massive genital bleeding
Medical history: Chronic myelogenous leukemia (41 years of age)
Family history: None

Menstrual history: First menstruation at the age of 13 years. She had hypermenorrhea, menstrual blood clots, and severe dysmenorrhea.

Present illness: She developed hypermenorrhea five years ago. At the age of 41 years, she was diagnosed with chronic myelogenous leukemia. Therefore, dasatinib (a tyrosine kinase inhibitor) was initiated. Consequently, the platelet count reduced, showing worsening hypermenorrhea. During menstruation, she visited our department for massive genital bleeding. This patient had a desire of having children.

Physical examination findings: There were no notable findings.

Blood test findings: blood test findings are shown in Table 1. Severe anemia and decreased platelet count were observed (hemoglobin, 2.7 g/dL; platelet, 1.6 million/μL).

**Table 1.** Blood test findings.

| White Blood Cell (WBC) | 1270/μL | Total Protein (TP) | 4.8 g/dL |
|---|---|---|---|
| Red blood cell (RBC) | 131 million/μL | Albumin (Alb) | 3.1 g/dL |
| Hemoglobin (Hb) | 2.7 g/dL | Total bilirubin (T.bil) | 0.5 mg/dL |
| Hematocrit (Ht) | 8.00% | Aspartate aminotransferase (AST) | 20 U/L |
| Platelet (Plt) | 1.6 million/μL | Alanine aminotransferase (ALT) | 15 IU/L |
| | | Blood urea nitrogen (BUN) | 12.2 mg/dL |
| Prothrombin time and international normalized ratio (PT/INR) | 1.09 | Creatinine (Cre) | 0.5 mg/dL |
| Activated partial thromboplastin time (aPTT) | 26.6 s | Lactate dehydrogenase (LDH) | 83 IU/L |
| | | Alkaline phosphatase (ALP) | 35 IU/L |
| | | Creatine kinase (CK) | 79 IU/L |
| | | Sodium (Na) | 138 mEq/L |
| | | Potassium (K) | 3.6 mEq/L |
| | | Chloride (Cl) | 108 mEq/L |

Severe anemia and decreased platelet count were observed.

Transvaginal ultrasound findings: A 65-mm mass was detected on the anterior uterine wall. No notable findings were observed in the bilateral uterine appendages.

Pelvic MRI findings: The sagittal plane of the T2-weighted images showed a 65-mm low-signal-intensity mass in the anterior uterine wall (shown in Figure 1a).

Cervical cytology: There were no special findings in the preoperative cervical and endometrial cytology results.

Treatment course: Based on the aforementioned findings, she was diagnosed with hypermenorrhea due to myoma or drugs. Dasatinib was discontinued, and she received a transfusion of RBCs and Plts. To control hypermenorrhea, microwave ablation of the inflowing blood vessels to the uterine myoma under transvaginal ultrasound guidance was planned after obtaining written informed consent from the patient.

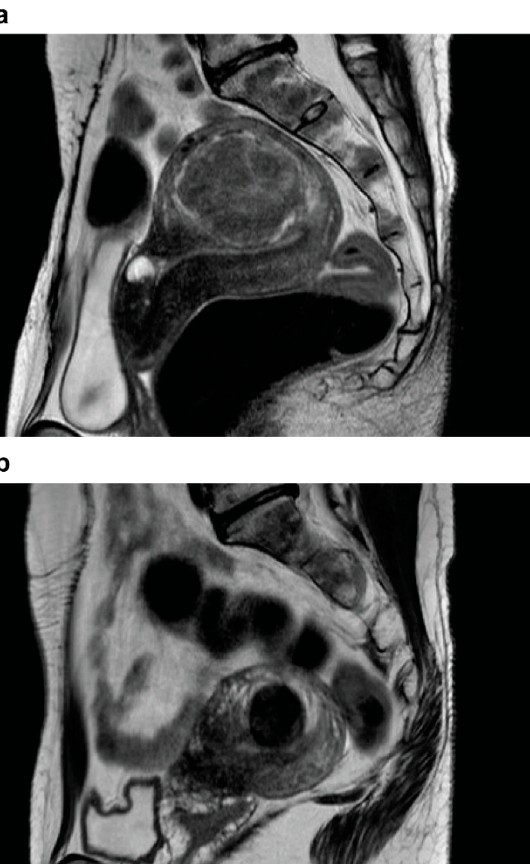

**Figure 1.** T2-weighted images of pelvic MRI (sagittal plane): (**a**) before surgery; (**b**) 3 months after surgery. The myoma volume was reduced from 65 mm (before surgery) to 27 mm (3 months after surgery).

Surgical findings: The patient underwent surgery in the lithotomy position with intravenous anesthesia. Microwave ablation of the inflowing blood vessels to the uterine myoma under transvaginal ultrasound guidance was performed using a Microtaze AFM-712 (Alfresa Pharma Co., Osaka, Japan) and a coagulation needle electrode (CB-type, CMD-16CBL-10/350) with a diameter of 1.6 mm and a length of 350 mm (Alfresa Pharma Co., Osaka, Japan) (shown in Figure 2). The schema of the procedure is shown in Figure 3. After locating the inflowing vessels of the myoma under transvaginal ultrasound using a color Doppler method (shown Figure 4a), direct ablation of these vessels was performed using 2.45 GHz microwave with a needle electrode (shown in Figure 4b). Ablation per lesion was performed using the following condition: power output of the Microtaze of 30 W and ablation duration of 10 s × five sets. Ablation was applied to five lesions in the feeding vessels around the myoma in the patient. The surgical time was 45 min. There was a small amount of bleeding. The postoperative course was good. Therefore, she was discharged at 4 h after surgery and followed up in an outpatient clinic.

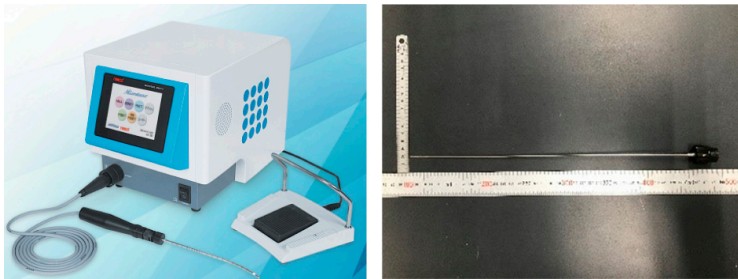

**Figure 2.** Microtaze AFM-712 generator and a coagulation needle electrode (Alfresa Pharma Co., Osaka, Japan).

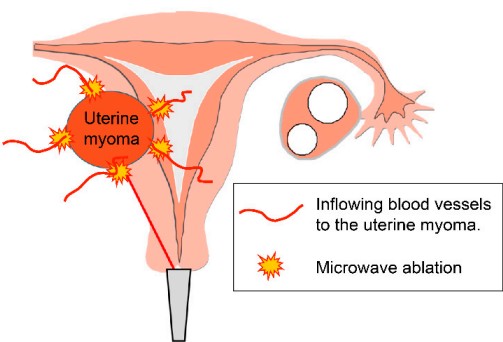

**Figure 3.** Schema of the procedure. After locating the inflowing vessels of the myoma under transvaginal ultrasound using a color Doppler method, direct ablation of the feeding vessels of the myoma was performed using 2.45 GHz microwave with a needle electrode.

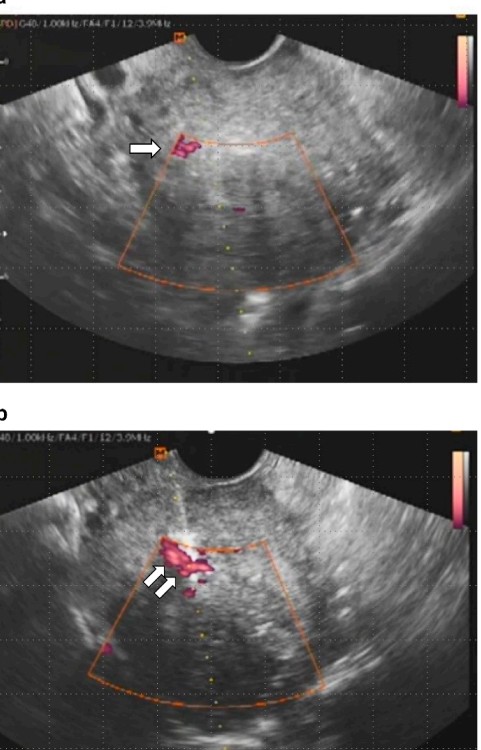

**Figure 4.** Intraoperative transvaginal ultrasound: (**a**) After locating the inflowing vessels of the myoma under transvaginal ultrasound using a color Doppler method (⇒); (**b**) microwave ablation of the inflowing feeding vessels of uterine fibroids using a needle-shaped deep coagulation electrode under transvaginal ultrasound guidance can be confirmed as a hyperechoic area. (⇒⇒).

Microwave ablation of the inflowing blood vessels to the uterine myoma under transvaginal ultrasound guidance was performed using a Microtaze AFM-712 and a coagulation needle electrode with a diameter of 1.6 mm and a length of 350 mm.

Subjective assessment using the visual analog scale showed a considerable improvement in clinical symptoms.

HB level increased significantly from 2.7 g/dL before surgery to 13.2 g/dL after surgery.

Postoperative course: Menstruation resumed at 45 days after surgery. Subjective assessment using the visual analog scale showed a considerable improvement in clinical symptoms (the score for hypermenorrhea improved from 10 [before surgery] to 1 [after surgery]; that for algomenorrhea improved from 10 [before surgery] to 2 [after surgery]) (shown in Figure 5). The Hb value considerably increased to 12.3 g/dL (shown in Figure 6). In addition, the myoma size reduced from 65 mm [before surgery] to 27 mm [3 months after surgery] (shown in Figure 1b). No complications were noted during the course. No hypermenorrhea recurrence was observed at 18 months after surgery. This patient is currently undergoing infertility treatment.

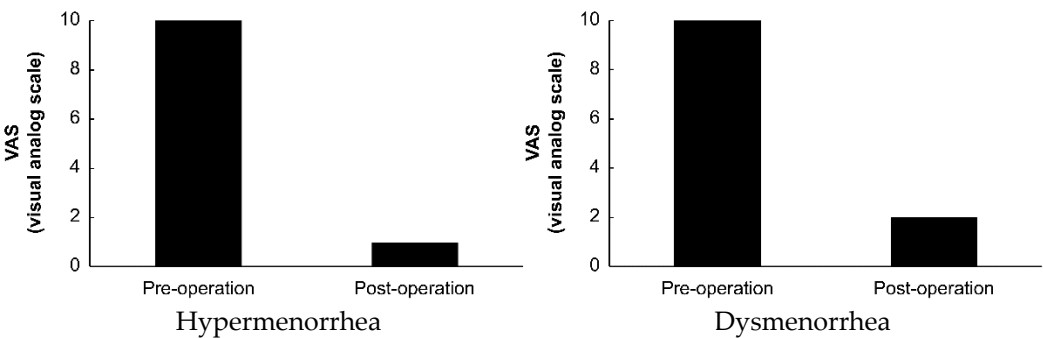

**Figure 5.** Evaluation of menorrhagia and dysmenorrhea before and after surgery using a visual analog scale.

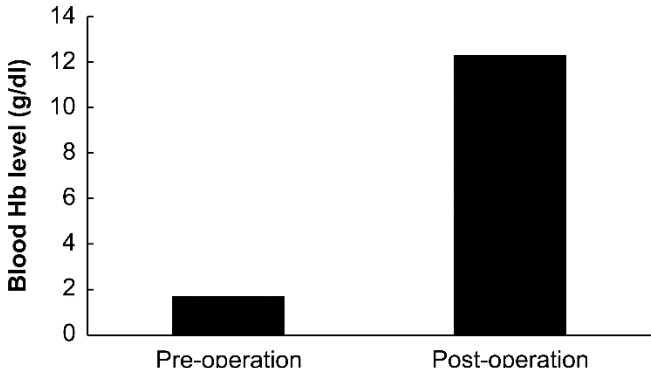

**Figure 6.** Changes in hemoglobin (Hb) levels.

### 3. Discussion

MEA, which includes 2.45 GHz microwave irradiation and ablation of the endometrial basal layer, reportedly has clinical benefits as a substitution therapy so that total hysterectomy for hypermenorrhea can be avoided [1,6,7]. MEA may have therapeutic effects on uterine myoma. Although MEA can be expected to have a therapeutic effect on uterine fibroids, there are some cases in which additional treatment is required due to postoperative recurrence [4,5] This may be caused by a dilated uterine cavity due to enlarged myomas after MEA, resulting in endometrial regeneration over time. UAE, F MRgFUS, and transcervical microwave myolysis (TCMM) are available as uterine-sparing treatments for uterine fibroids associated with menorrhagia that can be performed in Japan.

UAE is an interventional radiology approach for transcatheter embolization of the uterine artery. Since Ravina in France reported about UAE being used as a treatment for uterine myoma in 1995 [8], it has drawn attention as a minimally invasive alternative treatment for total hysterectomy. UAE has been widely used worldwide, mainly as a treatment for symptomatic uterine myoma. UAE has been reported to be equally effective in improving clinical symptoms, such as hypermenorrhea, and patient satisfaction compared with operative management, which includes total hysterectomy and myomectomies [9,10]. In contrast, the most common complication of UAE is postoperative fever, followed by pain, endometritis, uterine adhesions, and uterine necrosis [10,11]. In addition, regarding reproductive function following the UAE [12], ovarian dysfunction has been reported to not be observed, and it is also used for women who wish to become pregnant. However, other complications of UAE that may affect fertility include ovarian failure and secondary amenorrhea caused by endometrium atrophy and uterine cavity adhesions [9,13]. The possible effects of UAE on the uterine adnexa may include ovarian failure caused by reduced blood flow to the ovaries, infection-related damage to the fallopian tubes, and infertility due to these factors. There are some studies on pregnancy and birth after UAE [14,15]. However, some studies on UAE reported a higher incidence of miscarriage [14] and placental abnormalities, such as placenta accreta [15], after UAE. Therefore, pregnant women after UAE require strict perinatal management.

MRgFUS is a conservative treatment for uterine myoma. In this method, high-power ultrasound, also referred to as high-intensity focused ultrasound, is converted into thermal energy, leading to coagulation necrosis in the focal area. MRgFUS in combination with real-time MRI is a noninvasive extracorporeal thermal ablation method for the treatment of myoma. Reduced myoma volume and improved clinical symptoms, such as hypermenorrhea, have been reported [16]. However, irradiation of a large myoma is time-consuming. Additionally, reduced effects of ablation in patients with obesity and degenerated myoma may lead to an increase in the size of myoma and relapse of clinical symptoms [16]. In Japan, this therapy is not indicated for patients who wish to have children.

Similar to the development of our treatment, using microwaves, TCMM can be used for ablation of the uterine myoma. TCMM, including MEA for uterine myoma itself, uses a needle-type electrode compatible with a sounding applicator for MEA, which was developed based on conventional electrodes for endometrial ablation. The clinical effects of TCMM, including improved hypermenorrhea and anemia and a reduction in myoma size, have been reported [17–19]. However, the ablation of a large myoma is time-consuming because the range of ablation is approximately 6 mm from the surface of the sounding applicator. In addition, owing to the large diameter (4 mm) of a sounding applicator used for ablation, TCMM may cause surgical stress. Furthermore, because no study has examined the therapeutic effects of TCMM alone, the method that contributed to the success of this intervention is unknown.

Thus, in this study, we performed a treatment involving microwave ablation of the inflowing blood vessels to the uterine myoma under transvaginal ultrasound guidance. In this method, the location of the inflowing blood vessels of the myoma was identified prior to ablation using the color Doppler method. Then, microwave ablation of the selected inflowing blood vessels was performed under ultrasound guidance. This treatment may have contributed to the reduction in surgical time. Microwave ablation of the the inflowing blood vessels vessels of the myoma not only reduced blood flow but also improved clinical symptoms, such as hypermenorrhea and anemia, thereby resulting in a reduction in myoma size. In addition, the narrow coagulation needle electrode (1.6 mm diameter) used for the ablation of the inflowing blood vessels of the myoma may be useful for minimally invasive surgery.

Conventional MEA, which includes ablation of the endometrium (including basal layer) to reduce the amount of menstruation by preventing cyclic endometrial regeneration, is not indicated for the treatment of women wishing to conceive. MRgFUS and TCMM, which are conservative treatments for uterine myoma, are also not indicated for the treat-

ment of women wishing to conceive. Furthermore, UAE reduces blood flow in some cases, which may have a negative effect on the uterus and ovaries. Therefore, caution should be exercised when considering ablation in women wishing to conceive. The method used in this study included direct microwave ablation of the inflowing blood vessels of the myoma without ablation of the endometrium. Thus, this method may minimize the reduction in blood flow to the uterus and ovaries. Improved clinical symptoms in the present study suggest the clinical utility of our method as a conservative treatment for women with uterine myoma who prefer fertility preservation. Recent lifestyle changes among women have resulted in a significant trend of late marriages and pregnancies. Therefore, in several cases, symptomatic uterine myoma in women of reproductive age requires not only minimally invasive treatment but also management focusing on fertility preservation.

Further studies with a larger sample size and long-term follow-up of cases with uterine myoma with hypermenorrhea who underwent the procedure should examine the effects of such treatments from various perspectives, such as clinical efficacy, safety, and postoperative changes in hormone dynamics.

Microwave ablation of the inflowing blood vessels to the uterine myoma under transvaginal ultrasound guidance showed efficacy and safety, which was comparable to those exhibited by conventional MEA. Furthermore, this method also reduced the myoma size. These findings suggest that this method is a novel treatment option for patients with myoma-related hypermenorrhea who wish to preserve their fertility and have children.

**Author Contributions:** Conceptualization, T.K.; data curation, T.K., T.O., T.S., A.S., R.O., M.K., A.K., K.K., K.Y., N.T. and M.O.; methodology, T.K.; investigation, T.K., T.O., T.S., A.S., R.O., M.K., A.K., K.K., K.Y., N.T. and M.O.; formal analysis, T.K.; writing—original draft preparation, T.K.; writing—review and editing, T.K. All authors have read and agreed to the published version of the manuscript.

**Funding:** This research received no external funding.

**Institutional Review Board Statement:** Study approval statement: this study is approved by the ethics committee of the International University of Health and Welfare Hospital Approval Number: referral number: 20-B-399.

**Informed Consent Statement:** Informed and written consent was obtained from all subjects involved in the study.

**Data Availability Statement:** The datasets generated during and/or analyzed during the current study are available from the corresponding author on reasonable request.

**Conflicts of Interest:** The authors declare no conflict of interest.

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
