# Peer review of "Myoma with Hypermenorrhea Treated with Ultrasound-Guided Microwave Ablation of the Inflowing Blood Vessels to the Uterine Myoma: A Case"

_endocrines, doi:10.3390/endocrines3040054_

Round 1
Reviewer 1 Report
The aim of this case report is to describe the transvaginal ultrasound-guided microwave ablation as a new treatment for symptomatic uterine fibroids. The chosen case is representative of a condition of urgency (anemia due to hypoplateletpenia / menorrhagia) managed in a conservative manner and in respect of the possible future desire to conceive. Although the study design is appropriate, there are many areas of weakness and methodological inaccuracies:
1. The references cited are not recent, most of them are prior to 2017 and do not take into consideration the most used myolysis technique which involves the use of radiofrequency.
2. Menorrhagia should preferably be quantified with a pictorial blood loss assessment chart (PBAC) scoring system rather than with a subjective assessment as the visual analog scale.
3. The description of the electrode needle and the coagulation modalities are inadequate for a sientific report. The electrodes used must be characterized in their length and diameter with units of measurement respectively of centimeters and Gauge. The ablation quantication must be reported with the amount of total energy delivered to the tissue (unit of measurement Watt) in addition to the time of use.
4. The data are not interpreted appropriately and consistently throughout the manuscript. Uterine fibroid, for example, is never shown on clear ultrasound scans. Furthermore, the magnetic resonance images do not contain myoma measurements which are only reported in the legend. Figure 2 is poor in the details of the devices and the design needs to be definitely improved. Figure 3 does not clearly clarify the precision of intervention in vascularized areas.
5. Furthermore, considering the report of a single case, it is necessary to have a longer follow-up (6 or better 12 months) to be able to sustain the "EFFICACY" (line 208) of the proposed technique.
6. The discussion must be completely revised by giving less emphasis to Microwave Endometrial Ablation (MEA) and by citing more comparison with other emerging mini-invasive methods such as myolysis with radiofrequency or other experiences with microwaves.
Author Response
Please find the responses attached.

Reviewer 2 Report
The idea of the study is interesting and seems as probably considerable treatment option for some patients.
I would think about reformulation of the manuscript title - should be shorter and more precise. E.g. The microwave myoma ablation as fertility preserving procedure.
In the introduction destructive options as UAE, MRgFUS, MEA, TCMM should be explained in detail. In discussion their weak and strong points should be described as well as methods should be compared.
In such a study strongest interest of readers is associated with clinical applicability, so comparison of the methods might be placed in the table, e.g. strong/weak sides; inclusion and exclusions; current opinions, studies, research…
Analysis of HGB after 3 mths without other blood parameters e.g. HCT is useless. Moreover I would really expect longer perspective of observation. At least 6 mths optimally one year follow up, however I understand study limitations.
Figure 4. The US image is of a low quality, resolution of the figures should be better.
There are minor editorial and misleading usages of words/phrases which should be fixed prior publication.
I would expect more comprehensive discussion as mentioned above + refresh of the references (more up to date positions).
Author Response
Please find the responses attached

Reviewer 3 Report
I am very pleased you let me the opportunity to revise this interesting manuscript. I found the argument well targeted to the journal. However, in my opinion, the manuscript requires major revisions. Following here are reported my points.
A) Please follow the indications of the CARE statement while reviewing the manuscript. [Gagnier JJ, Kienle G, Altman DG, Moher D, Sox H, Riley D; CARE Group*. The CARE Guidelines: Consensus-based Clinical Case Reporting Guideline Development. Glob Adv Health Med. 2013 Sep;2(5):38-43. doi: 10.7453/gahmj.2013.008. PMID: 24416692; PMCID: PMC3833570.] AND [Riley DS, Barber MS, Kienle GS, Aronson JK, von Schoen-Angerer T, Tugwell P, Kiene H, Helfand M, Altman DG, Sox H, Werthmann PG, Moher D, Rison RA, Shamseer L, Koch CA, Sun GH, Hanaway P, Sudak NL, Kaszkin-Bettag M, Carpenter JE, Gagnier JJ. CARE guidelines for case reports: explanation and elaboration document. J Clin Epidemiol. 2017 Sep;89:218-235. doi: 10.1016/j.jclinepi.2017.04.026. Epub 2017 May 18. PMID: 28529185.]
B) The title should clearly reflect that this is a case report.
C) The English language of the abstract should be revised for clarity. And it should better reflect the CARE checklist requirements.
D) Most of the information reported in the first part of the case presentation should be tabulated and removed from the text for clarity.
E) I also suggest drawing a timeline or creating a table to include the repeated examinations and tests.
F) In my opinion, to assess the vascularization should be considered in the discussion section also the use of the microbubble contrast agents.
G) A revision of the literature should also be included considering similar techniques presented by other authors (e.g., Kanaoka Y, Yoshida C, Fukuda T, Kajitani K, Ishiko O. Transcervical microwave myolysis for uterine myomas assisted by transvaginal ultrasonic guidance. J Obstet Gynaecol Res. 2009 Feb;35(1):145-51. doi: 10.1111/j.1447-0756.2008.00872.x. PMID: 19215562.)
Author Response
Please find the responses attached

Round 2
Reviewer 3 Report
The manuscript is significantly improved and suitable to be published.